# Antiulcer Activity of Steamed Ginger Extract against Ethanol/HCl-Induced Gastric Mucosal Injury in Rats

**DOI:** 10.3390/molecules25204663

**Published:** 2020-10-13

**Authors:** Jun-Kyu Shin, Jae Hyeon Park, Kyeong Seok Kim, Tong Ho Kang, Hyung Sik Kim

**Affiliations:** 1School of Pharmacy, Sungkyunkwan University, Seobu-ro, Suwon 16419, Gyeonggi-do, Korea; allzzangjk@naver.com (J.-K.S.); sky3640@naver.com (J.H.P.); caion123@nate.com (K.S.K.); 2Department of Oriental Medicine Biotechnology, Graduate School of Biotechnology, Kyung Hee University, Yongin 17104, Gyeonggi-do, Korea

**Keywords:** *Zingiber officianale*, steamed ginger extract, antiulcer, oxidative stress, pro-inflammatory cytokines

## Abstract

Ginger (*Zingiber officianale*), the most widely consumed species, is traditionally used as a folk medicine to treat some inflammatory diseases in China and Korea. However, the functional activity of steamed ginger extract on gastric ulcers has not been previously explored. The present study aimed to investigate antiulcer activity of steamed ginger extract (GGE03) against ethanol (EtOH)/HCl-induced gastric ulcers in a rat model. GGE03 (100 mg/kg) was orally administered for 14 days to rats before oral intubation of an EtOH/HCl mixture to induce gastric damage. Pretreatment with GGE03 markedly protected the formation of microscopic pathological damage in the gastric mucosa. Further, administration of GGE03 significantly increased mucosal total nitrate/nitrite production in gastric tissues, and elevated total GSH content, catalase activity and superoxide dismutase (SOD) expression as well as decreasing lipid peroxidation and myeloperoxidase (MPO) activity. Underlying protective mechanisms were examined by assessing inflammation-related genes, including nuclear factor-κB (NF-κB), prostaglandin E2 (PGE_2_), and pro-inflammatory cytokines levels. GGE03 administration significantly reduced the expression of NF-κB and pro-inflammatory cytokines. Our findings suggest that GGE03 possesses antiulcer activity by attenuating oxidative stress and inflammatory responses.

## 1. Introduction

Gastric ulcers are one of the most prevalent types of peptic ulcer commonly seen in humans, affecting more than 10% of the global population [1]. The principal etiological factors associated with gastric ulcers are alcohol and non-steroidal anti-inflammatory drugs (NSAIDs) abuse, stress, smoking, and infection of *H. pylori* [2,3,4,5]. Among risk factors, alcohol consumption can directly interfere with gastric motility and metabolism. This action leads to mucosal damage and ulceration in the stomach [6]. Increasing evidence shows that an ethanol-induced gastric ulcer is closely associated with the generation of reactive oxygen species (ROS). Overproduction of ROS under oxidative stress results in cellular damage in the stomach. Therefore, gastric cells exhibit induction of several endogenous antioxidant enzymes, such as superoxide dismutase (SOD), glutathione peroxidase (GPx), and catalase (CAT) to maintain gastrointestinal homeostasis through the scavenging of ROS [7]. Further, alcohol-mediated ROS is a crucial factor in inflammation via an increasing formation of pro-inflammatory cytokines, such as tumor necrosis factor-α (TNF-α) and interleukin-1β (IL–1β), which exacerbate inflammation [8].

Gastric ulceration occurs when the imbalance between causative factors (acid and pepsin or *H. pylori* infection) and defensive factors (prostaglandins (PGs), nitric oxide (NO), bicarbonate, and mucin) [9,10]. The primary drugs used to minimize ulcer symptoms are H2 receptor antagonists, proton pump inhibitors, and antacids. However, these drugs have side effects, such as gynecomastia, vitamin B_12_ hypergastrinemia and hypercalcemia [11,12,13]. Mucosal protective agents with relatively low side effects are beneficial alternatives, and some medicinal herbs have been used for the treatment of gastric ulcers [14].

Ginger (*Zingiber officianale*) is an herbaceous perennial plant in the Zingiberaceae family. It is widely distributed and used in Asian countries as a traditional medicine for a wide array of conditions, including nausea, vomiting, colds, fever, and rheumatic disorders [15]. Several studies report that ginger and its active components have antimicrobial [16], antioxidant [17], and anti-inflammatory properties [18]. Ginger extract showed its protective effects by reducing lipid peroxidation and enhancing antioxidant enzyme activities in mycotoxin-treated HepG2 cell lines [19]. Supplementation with ginger powder for ten weeks decreased the nuclear factor kappa light chain enhancer of activated B cell (NF-κB) expression in peripheral blood mononuclear cells in patients with type 2 diabetes [20]. Ginger extract and its active compounds also show anti-ulcerative activity in several acute or chronic gastric ulcer animal models through a downregulation of inflammatory cytokine release and an upregulation of antioxidant enzyme activity [21,22,23].

Processing by steaming affects the chemical profiles of natural products and leads to changes in bioactivity [24]. Steaming of ginseng alters its composition, and increases the chemo-preventive effects against various cancer models [25,26]. Further, steamed ginger shows both lower toxicity and greater efficacy than ginger extract, and demonstrates anti-hyperglycemic activity in diabetic mice [27]. However, no research on the protective effects of steamed ginger is available from animal models of gastric ulcers.

To the best of our knowledge, this study is the first to investigate the protective effects of steamed ginger extract (GGE03) against gastric mucosal damage in rats. An ethanol-induced gastric ulcer model has been commonly used to study both pathogenesis and therapies for the human ulcerative disease. In the present study, we investigated the protective effect and its mechanisms of action of GGE03 on gastric ulcer in an acidified ethanol-induced gastric ulcer animal model.

## 2. Results

### 2.1. Macroscopic and Histological Effect of GGE03 on EtOH/HCl-Induced Gastric Mucosal Injury

No significant differences in body weight were exhibited among experimental groups (GGE03 100, and 300 mg/kg) and the EtOH/HCl group during the experimental period (data not shown). Intragastric administration of EtOH/HCl resulted in multiple hemorrhagic lesions in the glandular portion of the stomach, with a lesion index of 95.6 ± 7.9 mm^2^ (Figure 1a–g). Treatment with GGE03 (100 and 300 mg/kg) resulted in a significant reduction in gastric ulcer lesions, with an inhibition of 34% and 28%, respectively. However, administration of GGE03 at doses of 10 and 30 mg/kg did not show gastroprotective effects (Figure 1h).

Thus, we selected a GGE03 dose of 100 mg/kg an optimal effective dose for further investigation of EtOH/HCl-induced acute gastric ulcers. Finally, animals that received indomethacin (100 mg/kg, positive control) also showed a significant reduction in gastric ulcer lesions, with 91.0% ulcer inhibition.

Histological analysis showed that acidified ethanol extensively disrupted the superficial region of the gastric gland, causing epithelial cell loss, edema, and intense hemorrhage in the glandular portion of the stomach (Figure 2). Sham + Vehicle group rats did not show gastric mucosal structure and stomach glandular structural loss, hemorrhage, or submucosal edema (Figure 2a). In contrast, extensive stomach glandular structural loss, edema, leukocyte infiltration, confluent necrosis, and serious hemorrhaging were observed in animals treated with EtOH/HCl (Figure 2c). Administration of GGE03 (100 mg/kg) effectively ameliorated gastric injury caused by EtOH/HCl (Figure 2d). Histomorphometric analysis (Figure 2e) showed significant (*p* < 0.05) increases in the total histological scores of gastric ulceration in the EtOH/HCl group compared to the Sham + Vehicle group rats; however, these scores were significantly lower after pretreatment with GGE03 (Table 1).

### 2.2. Effect of GGE03 on Mucosal Defensive Factors in EtOH/HCl-Treated Rats

Rats treated with acidified EtOH showed a significant decrease in the gastric wall mucus (6.4 ± 0.4 μg/mg tissue) compared to the vehicle-treated sham group (9.9 ± 0.1 μg/mg of tissue). Administration of GGE03 (100 mg/kg) significantly attenuated the development of acute gastric ulcers (Figure 3a). Gastric NO and PGE_2_ play key roles in the maintenance of gastrointestinal mucosa [28]. Gastric PGE_2_ levels in vehicle-treated sham group rats were 36.9 ± 1.7 ng/mg of tissue (Figure 3b). Administration of acidified ethanol significantly reduced PGE_2_ levels to 29.0 ± 1.0 ng/mg of tissue. This reduction was attenuated by pretreatment with 100 mg/kg GGE03 (39.4 ± 0.9 ng/mg of tissue). Further, total nitrate/nitrite content was 15.4 ± 1.4 μM in the vehicle-treated sham group, administration of acidified ethanol decreased gastric NO levels to 6.8 ± 0.8 μM, and GGE03 pretreatment (100 mg/kg) significantly restored NO to control levels (Figure 3c).

### 2.3. Effect of GGE03 on Antioxidants Activities in EtOH/HCl-Treated Rats

We measured gastric malondialdehyde (MDA), antioxidative enzyme activities, and protein expression to evaluate the antioxidative properties of GGE03. MDA is globally accepted as a marker of metabolic oxidative stress during inflammation [29]. Gastric MDA levels in sham + vehicle group animals were 0.4 ± 0.1 nmol/g protein (Figure 4a). Treatment with EtOH/HCl significantly increased this level to 1.2 ± 0.1 nmol/g protein; this increase was attenuated by GGE03 treatment. Glutathione (GSH), catalase (CAT) and SOD are important for enzymatic responses to oxidative stress [30]. Intragastric administration of EtOH/HCl significantly decreased GSH content and CAT and SOD activity in comparison with vehicle-treated sham group animals (Figure 4b–d). These changes were attenuated by GGE03.

### 2.4. Effect of GGE03 on Anti-Inflammatory Properties in EtOH/HCl-Treated Rats

Myeloperoxidase (MPO) is an enzyme abundantly expressed in neutrophil granulocytes and serves as a predictor of inflammation for several diseases. In vehicle-treated sham group rats, the MPO level was 0.38 ± 0.07 U/mg of tissue. EtOH/HCl treatment significantly increased MPO to 0.69 ± 0.02 U/mg of tissue. This increase was attenuated by GGE03 treatment (Figure 5a).

Protein expression related to inflammation signaling was evaluated (Figure 5b) to verify the mechanisms of gastroprotective effects of GGE03. NF-κB is responsible for controlling inflammation signaling in many inflammatory diseases. In vehicle-treated rats, nuclear translocation of NF-κB was barely detected. EtOH/HCl treatment resulted in a significant increase in the nuclear translocation of NF-κB in comparison with that of the vehicle-treated sham group. Further, cytosolic IkB-α protein expression was significantly decreased in comparison with that of controls. Pretreatment with GGE03 attenuated these changes. Moreover, we evaluated pro-inflammatory cytokine mRNA and protein expression associated with NF-κB signaling. IL-1β and TNF-α mRNA expression were significantly increased by EtOH/HCl treatment compared with sham + vehicle group rats (Figure 5c). Protein expression of IL-1β and TNF-α seemed to show the same pattern as mRNA expression, and GGE03 pretreatment again attenuated changes in expression (Figure 5d).

## 3. Discussion

Ethanol-mediated gastrointestinal disorders affect millions of people and are a major global health concern. Ethanol stimulates the gastric mucosal H^+^/K^+^-ATP pump, which leads to the secretion of gastric acid and pepsin. This action also blocks the blood flow, and K^+^ and Na^+^ pumps that cause gastric acid leaping [31]. Ethanol accelerates gastric injury through recruitment of immune cell infiltration that initiates an inflammatory cascade; HCl increases oxidative stress and corrosive damage to gastric mucus [32]. Thus, we used acidified ethanol to produce ulceration that resembles human gastric ulcer in a commonly used animal model [33]. Our studies highlight the gastroprotective effects of steamed ginger (GGE03) on acidified ethanol-induced acute gastric ulcers both directly via inhibition of inflammatory responses and indirectly through increasing antioxidant enzyme capacity.

Increases in PGE_2_ and NO levels are likely key protective mechanisms against gastric mucosal damage [34,35]. PGE_2_, as an endogenous gastroprotective factor, plays an important role in gastric mucosal defense by improving blood flow, stimulating mucus and bicarbonate secretion, and strengthening the resistance of epithelial cells to stimuli [36]. PGE_2_ biosynthesis is mediated by cyclooxygenase (COX) enzymes. COX consisted of COX-1, a constitutive form, and COX-2, a form induced under pathophysiological situations such as inflammation [37]. Decreased production of PGE_2_ by COX-1 inhibition is an important factor in the pathogenesis of gastric ulcers [38]; however, some studies have demonstrated that COX-2 inhibition showed beneficial effects on gastric ulcers [39]. Ginger extract suppressed inflammatory responses by downregulation of COX-2/PGE_2_ overproduction in lipopolysaccharide-treated Caco-2 cell lines [40].

NO, a powerful vasodilator, regulates bicarbonate secretion, blood flow, and neutrophil aggregation in the stomach [41]. Like PGE_2_, NO derived from nitric oxide synthase (NOS) has diverse effects on physiological and pathophysiological conditions. Recently, Chatterjee et al. [42] demonstrated that an increase of endothelial NOS (eNOS) is key to ulcer healing from indomethacin-mediated gastritis. Further, oral administration of 6-shogaol, a representative active component of ginger, showed protective effects by regulating pro-inflammatory factors such as iNOS, TNF-α and IL-1β in dextran sulfate sodium-induced colitis [43] and aspirin-induced gastric ulcer [44]. In our study, acidified ethanol inhibited gastric PGE_2_ and NO levels and GGE03 attenuated these decreases. These results suggest that GGE03 treatment alleviated EtOH/HCl-induced gastric injury by increasing gastric defense factors.

Infiltration of neutrophils into the gastric mucosa plays an important role in the pathogenesis of gastric mucosal inflammation and damage [45]. Migrating neutrophils and macrophages release several pro-inflammatory cytokines and ROS during gastric inflammation [46]. MPO, a peroxidase enzyme, produces hypochlorous acid from hydrogen peroxide during inflammation-mediated neutrophil bursts. Thus, increased MPO is used as a biomarker of neutrophil infiltration into damaged tissue. Acidified ethanol in the present study significantly increased MPO activity in gastric tissue, indicative of gastric inflammation. Treatment with GGE03 significantly attenuated this increase in MPO activity. This finding is also supported by histological analysis of the gastric mucosa. Microscopic examination revealed gastric edema, loss of epithelial cells and severe hemorrhage following development of EtOH/HCl-induced gastric ulcers. GGE03 ameliorated these histological changes in the gastric tissue. Further, TNF-α is a well-known pro-inflammatory cytokine that enhances the production of IL-1β and IL-6 and induces the activation of NF-κB through binding to the TNF-receptor [47]. NF-κB, a heterodimeric transcription factor, mediates transcription of an array of proteins involved in inflammatory responses and apoptotic cell death [48]. While in an inactivated state, NF-κB is located in the cytosol as a complex with inhibitory protein IκB. In a pathological condition, such as inflammation, IκB protein is phosphorylated or ubiquitinated and finally results in degradation and activation of NF-κB translocation into the nucleus. Translocated NF-κB triggers transcriptional activation of pro-inflammatory cytokines [49]. Several studies suggest that NF-κB activation is responsible for the pathogenesis of inflammatory diseases, and its modulation is of interest as a therapeutic target for gastric ulcers [50]. Lee et al. [51] demonstrated that 1-dehydro-10-gingerdione suppressed NF-κB-regulated gene expression through inhibiting the catalytic activity of cell-free IKKβ in lipopolysaccharides-treated macrophages. Moreover, 6-gingerol, a representative active component of ginger, showed anti-inflammatory properties by inhibiting TRAIL-induced NF-κB activation [52]. In our study, administration of acidified ethanol significantly increased NF-κB activation through translocation from cytosol to the nucleus accompanied by expression of pro-inflammatory cytokines. GGE03 pretreatment attenuated these changes, indicating that GGE03 might have an anti-inflammatory property.

Oxidative stress is responsible for damaging gastric mucosal morphology and loss of mucosal integrity caused by several aggressive factors, such as alcohol. Gastric mucosa has a high amount of a thiol-containing tripeptide (GSH), which is a crucial cellular non-enzymatic defense mechanism against oxidative stress [53]. BSO, an irreversible inhibitor of GSH biosynthesis, abolished the protective effect of natural products against cysteamine-induced duodenal ulcers [54]. Furthermore, enzymatic antioxidant enzymes SOD and CAT scavenge and regulate overall ROS to maintain physiological homeostasis. Patients with peptic ulcers and gastric cancer display significantly reduced levels of SOD and CAT levels [55]. In our study, administration of EtOH/HCl significantly increased MDA levels, accompanied by a decreased GSH content, CAT activity, and SOD protein expression. Steamed ginger (GGE03) reduced gastric lipid peroxidation and increased antioxidant properties including GSH levels, CAT activity, and SOD expression, indicating that its anti-ulcerogenic properties are partially based on antioxidant function.

## 4. Materials and Methods

### 4.1. Preparation of Steamed Ginger Extracts (Golden Ginger, GG)

*Zingiber officinale* Roscoe was purchased from Wanju-gun (Jeonrabuk-do, Korea) in April 2019, and was identified by Professor Se Chan Kang, Kyung Hee University (Yongin, Gyeonggi-do, Korea). A voucher specimen (No. 713) was deposited in the Laboratory of Natural Medicine Resources at the BioMedical Research Institute, Kyung Hee University (Suwon, Korea). Ginger was washed three times with distilled water to remove sand and dust, and then steamed at: 2–2.5 kgf/cm^2^, 97 °C, for 2 h. Golden ginger (GGE) was obtained by extracting steamed ginger with fifteen-fold 70% ethanol (*v*/*v*) for 15 h at 85 °C, 1.5 kg/cm^2^. GGE was passed through a 60-mesh filter, concentrated at −650 mmHg, 55 °C. The extract (GGE03) was spray-dried to obtain a powder and stored at −20 °C until use (Scheme 1).

### 4.2. Quantitative Analysis of the Major Component of GGE

Calibration curves for each standard were made using six concentrations (3.125 to 100 μg/mL). GGE was filtered through 0.22 μm membrane filters (Woongki Science Co., Ltd., Seoul, Korea) and a 10 μL aliquot of each extract solution in 80% MeOH (10.0 mg/mL) was injected into the HPLC system. HPLC-grade acetonitrile and water were obtained from Honeywell Burdick and Jackson Inc. (Muskegon, MI, USA). HPLC analysis was achieved using a Waters 600S (Waters, Milford, MA, USA) and with a Waters 2487 UV detector (254 nm). The column was a Shimpack Gist (4.6 × 250 mm, particle size: 3 µm, Shimadzu Co., Kyoto, Japan). The mobile phase consisted of 0.1% formic acid in water (solvent A) and acetonitrile (solvent B), which were eluted at a flow rate of 0.4 mL/min with the following gradient elution with concentration of solvent from 30% (5 min) to 100% (60 min). The quantitative analysis was replicated three times. The content of 1-dehydro-6-gingerdione in the GGE was determined to be 0.19% [56].

### 4.3. Animal Treatment

Male Sprague–Dawley rats (6 weeks of age, 180–200 g) were purchased from Orient Bio Co. Ltd. (Gyeonggi-do, Korea). Animals were paired-housed (2 per cage) in a room with controlled temperature and humidity (25 ± 1 °C and 55 ± 5%, respectively) and with a 12 h light–dark cycle. Animals received a standard rodent chow ad libitum (Teklad Global 18% Protein Rodent Diet 2018S: Harlan Laboratories INC., Indianapolis, IN, USA).

Acute gastric ulcers were induced by EtOH/HCl treatment following a previous report [57]. In our preliminary experiment, after 24 h of fasting, rats were orally administered GGE03 (10, 30, 100, and 300 mg/kg), vehicle (normal saline) as a sham, or indomethacin (100 mg/kg) as a positive control, 1 h before receiving 1 mL of 150 mM HCl in 60% ethanol (p.o.). In the second experiment, rats were randomly divided into four groups (*n* = 6 for each group): vehicle-treated sham group (sham + vehicle), vehicle-treated GGE03 (sham + GGE03 100 mg/kg) group, vehicle-treated gastric ulcer model group (HCl/EtOH + vehicle), and gastric ulcer model with GGE03 (HCl/EtOH + GGE03 100 mg/kg) group. Vehicle control rats were treated with an equal volume of deionized water by oral gavage. Rats in the treatment group were orally administered 1 mL of test substances (GGE03) dissolved in distilled water consecutively for 14 days. On the last day of treatment, all animals were deprived of food for 24 h overnight in a cage. Rats were euthanized by an injection of ketamine and xylazine (55 mg/kg and 7 mg/kg, respectively, i.p.) and stomach tissues were removed. The stomachs from each group were opened along the greater curvature, and thoroughly rinsed with normal saline. A longitudinal section of the gastric tissue was taken from the anterior part of the stomach and fixed in 4% formalin solution for later histological analysis and the remaining stomach sections were stored at −75 °C for later biochemical analysis.

Animal experimental procedures were approved by the Sungkyunkwan University Animal Care Committee (SKKUIACUC2019-12-24-1) in accordance with National Institutes of Health Guidelines of the Care and Use of Laboratory Animals (NIH, Department of Health and Human Services Publication No. 85–23, revised 1985).

### 4.4. Macroscopic Assessment of Gastric Ulcers

At 24 h after induction of gastric lesions, rats were sacrificed, and their stomachs were immediately removed and opened along the greater curvature. Gastric damage (erosion or ulcer) was determined with “Image J” processing software (NIH, USA). The sum of lesion area (mm^2^) per stomach was used as a lesion index, and ulcer inhibition rate (%) was calculated as follows: Inhibition rate (%) = (1 − lesion index of test animal/lesion index of vehicle-treated EtOH/HCl animal) × 100.

### 4.5. Histological Analysis

After macroscopic analysis, a small portion of each stomach was fixed in 4% formalin solution for 24 h. Stomach tissue sections were dehydrated with graded concentrations of ethanol, passed through xylene, and embedded in paraffin. Paraffin sections (4 μm thick) were stained with hematoxylin and eosin. Histological evaluation was performed under a light microscope (Olympus PROVIS AX70; Tokyo, Japan). Histopathological changes were evaluated by a pathologist who was blinded to this study following a previously described method with slight modification [58].

### 4.6. Gastric Wall Mucus Contents

Gastric wall mucus content was determined as described by Kitagawa et al. [59] with slight modification. Briefly, resected stomachs were weighed for later assay and immersed in 0.1% (*w/v*) Alcian blue (Sigma Chemical, St. Louis, MO, USA) for 2 h. Excessive dye was removed by two successive rinses (15 min each) in 0.25 M sucrose solution. Mucus-bound dye was extracted with 10 mL of 0.5 M MgCl_2_ for 2 h with intermittent shaking every 30 min. Blue extract (800 μL) was shaken vigorously with an equal volume of diethyl ether and centrifuged at 500× *g* for 5 min. Optical density of the aqueous phase was measured using a spectrophotometer at a wavelength of 580 nm. The quantity of Alcian blue extract per gram of stomach was calculated from a standard curve.

### 4.7. Measurement of Gastric Mucosal NO and Enzyme-Linked Immunosorbent Assay for PGE_2_

A commercial nitrate/nitrite colorimetric assay kit (Cayman Chemical, Ann Arbor, MI, USA) was used for quantification of gastric mucosal NO content, and a PGE_2_ enzyme-linked immunosorbent assay kit (Cayman Chemical, Ann Arbor, MI, USA) was used for quantification of gastric mucosal PGE_2_ concentration.

### 4.8. Catalase Activity

Stomach tissue was homogenized in ice-cold PBS and centrifuged at 5000× *g* for 15 min. A commercial catalase colorimetric assay kit (Sigma-Aldrich) was used for quantification of the gastric catalase activity. Absorbance was read at 450 nm with a microplate reader.

### 4.9. Lipid Peroxidation and Glutathione Contents

The homogenates from gastric tissues were analyzed for malondialdehyde (MDA) by measuring the level of thiobarbituric acid (TBA)-reactive substances spectrophotometrically at 535 nm using 1,1,3,3-tetraethoxypropane (Sigma–Aldrich) as the standard [60]. Total GSH was determined in gastric homogenates using yeast-glutathione reductase, 5,5′-dithiobis(2-nitrobenzoic acid), and nicotinamide adenine dinucleotide phosphate after precipitation with 1% picric acid. Absorbance was read at 412 nm [61].

### 4.10. Measurement of Gastric MPO Activity

Myeloperoxidase (MPO) activity in gastric tissue was measured using a commercial MPO colorimetric activity assay kit (Sigma–Aldrich) following the manufacturer’s protocol.

### 4.11. Whole, Cytoplasmic, and Nuclear Protein Extraction of Stomach Tissue

Total protein was extracted using PRO-PREP Protein Extraction Solution (iNtRON Biotechnology Inc., Seongnam, Korea) in a micro-centrifuge tube. Briefly, isolated tissues were homogenized and incubated for 30 min in an ice-bath. Whole homogenates were centrifuged at 13,000× *g* for 6 min, 30 s, and total protein in supernatants determined using the BCA protein Assay kit (Pierce Biotechnology, Rockford, IL, USA).

We used NE-PER (Pierce Biotechnology) following the manufacturer’s instructions to isolate nuclear and cytoplasmic fractions. Briefly, stomach tissue was resected and homogenized with cold CERI (cytoplasmic extraction reagent I). After incubation for 10 min in an ice-bath, CERII was added and centrifuged at 16,000× *g* for 5 min. The supernatant was collected as a cytoplasmic extract, and we added a nuclear extract reagent to the nuclear pellet. After 40 min incubation in an ice-bath, we centrifuged again and collected the nuclear extract. Protein concentrations of cytoplasmic and nuclear fractions were measured using a BCA protein Assay kit (Pierce Biotechnology, Rockford, IL, USA).

### 4.12. Protein Extraction and Immunoblots

The quantitation of gastric protein used Western blotting. Gastrict issues were homogenized in Pro-prep reagent (17081; iNtRON Biotechnology, Seongnam, Korea) containing HaltTM phosphatase inhibitor cocktails (Thermo Scientific, Waltham, MA, USA) and protein concentrations were quantified using a PRO-MEASURETM kit (21011; iNtRON Biotechnology, Seongnam, Korea). After isolation, proteins (20 μg) were loaded onto 7.5%–10% polyacrylamide gels, separated by SDS/PAGE, and transferred to polyvinylidene fluoride membranes (Millipore, Billerical, MA, USA) using Semi-Dry Trans-Blot Cell (Biorad Laboratories, Hercules, CA, USA). After transfer, membranes were washed with 0.1% Tween-20 in 1× Tris-buffered saline (TBS/T) and blocked for 1 h at room temperature using a 5% (*w*/*v*) skim milk powder in TBS/T. Membranes were incubated for 16 h at 4 °C with the appropriate primary antibody. Membranes were then washed four times for 10 min each in TBS/T and incubated with appropriate secondary antibodies for 1 h at room temperature. Membranes were washed four times for 10 min each in TBS/T and examined using an ECL detection system (iNtRON Biotechnology Inc.). Band density was measured by TotalLab (TotalLab Ltd., Newcastle, UK). Primary antibodies against interleukin (IL)-1β (1:1000 dilution; Abcam, Cambridge, UK), IκB-α (1:1000; Santa Cruz Biotechnology, Santa Cruz, CA, USA), NF-κB (1:200 dilution; Santa Cruz), SOD (1:2500 dilution, Abcam), and TNF-α (1:200 dilution; Abcam), were used and signals were standardized to β-actin (1:5000 dilution, Sigma–Aldrich) for cytosolic and whole extracts and to lamin B1 (1:2000 dilution, Abcam) for nuclear fractions.

### 4.13. Determination of mRNA Extraction in Gastric Tissue

The gastric tissue was homogenized (Polytron PT 2500E, Kinematica AG, Luzern, Switzerland) and total RNA was extracted using Trizol following the manufacturer’s instructions (Invitrogen, Carlsbad, CA, USA). mRNA was then quantified with a spectrophotometer (BIOTEK, Inc., Winooski, VT, USA) to obtain 260/280 nm ratios. cDNA was synthesized by reverse transcription (EcoDry^TM^ cDNA Synthesis Premix, TaKaRa Bio Inc.). cDNA was then amplified by real-time PCR using a thermocycler (Lightcycler^®^ Nano, Roche Applied Science, Indianapolis, IN, USA) and an SYBR Green detection system (Roche Applied Science). Primer sequences are presented in Table 2. mRNA expression levels were normalized to levels of glyceraldehyde-3-phosphate dehydrogenase (GAPDH) and are expressed relative to the average of all delta cycle threshold (Ct) values in each sample using the Ct method. All experiments were conducted in duplicate to ensure amplification integrity.

### 4.14. Statistical Analysis

All results are presented as means ± standard error of the mean. Overall significance of results was examined by one-way ANOVA. Differences between groups were considered statistically significant at *p* < 0.05 with appropriate Bonferroni corrections made for multiple comparisons.

## 5. Conclusions

In conclusion, we elucidated the gastroprotective effects of GGE03, a steamed ginger preparation, against acute gastric ulcers in a rat model. Protective mechanisms appear to be (i) enhancement of gastric defensive mechanisms as evidenced by increased mucosal PGE_2_ synthesis and total nitric oxide, (ii) enhancement of gastric antioxidation capability, and (iii) inhibition of inflammation. The overall data support that GGE03 is a potential therapeutic agent for gastritis patients.

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
