# Peer review of "Antiulcer Activity of Steamed Ginger Extract against Ethanol/HCl-Induced Gastric Mucosal Injury in Rats"

_molecules, 2020, doi:10.3390/molecules25204663_

Round 1
Reviewer 1 Report
The manuscript titled “Antiulcer Activity of Steamed Ginger extract, GGE03, against Ethanol/HCl-induced Gastric Mucosal Injury in Rats” is focusing on the study of effects of the extract against formation of ulcer-like phenomena in the experimental in vivo model and its possible protective properties with a focus on antioxidant and anti-inflammatory properties.
These following concerns need to be addressed:
Specific comments:
- Citing item 1 (Liu E.S., Cho C. Relationship between ethanol induced gastritis and gastric ulcer formation in rats. Digestion 2000; 62: 232-239) as a source of information on the epidemiology of peptic ulcer disease is at least inappropriate and misleading. After all, it is devoted to one of the methods of conducting research that mimics the disease on experimental animals. Authors should place here the latest data on this disease from updated, as recent reports.
- A similar comment can be made to the introduction section devoted to oxidative stress and its relationship with peptic ulcer disease. The authors cite position 7 (Rezaie A., Parker R.D., Abdollahi M. Oxidative stress and pathogenesis of inflammatory bowel disease: An epiphenomenon or the cause? Dig Dis Sci . 2007; 52: 2015-2021) in it, which not only is a bit older, but also mainly refers to Crohn's disease and ulcerative colitis. At this point, Authors should cite works strictly devoted to the discussed topic.
- Also the citation of position 8 (Nian M., Lee P., Khaper N., Liu P. Inflammatory cytokines and postmyocardial infarction remodeling. Circ Res. 2004; 94: 1543-1553) is not very appropriate and does not fully reflect the idea presented in in the sentence (pages 1-2: "Further, alcohol-mediated ROS is a crucial factor in inflammation via increasing formation of pro-inflammatory cytokines, such as tumor necrosis factor-α (TNF-α) and interleukin-1β (IL-1β) , which exacerbate inflammation [8] ”), because to support this thesis there are many other reports dealing with the exact alcohol compound, ROS and the pro-inflammatory processes related to the release of various pro-inflammatory cytokines.
- It also seems that the citation of item 11 (Zakaria et al. Gastroprotective activity of chloroform extract of Muntingia calabura and Melastoma malabathricum leaves. Pharm Biol. 2016: 54: 812-826) in relation to the sentence (page 2, paragraph 2: “However, these drugs have side effects, such as gynecomastia, impotence, osteoporotic bone fracture, and deficiencies of iron and magnesium, as well as vitamin B12 hypergastrinemia after discontinuation”) discussing the adverse concerns about drugs H2 receptor antagonists, proton pump inhibitors and antacids is unjustified and not directly related to the idea contained in this fragment of the reviewed publication.
- Many parameters have been measured after inducing simulating ulcer formation in experimental animals (see: Materials and Methods chapter). It seems that Authors should describe exactly how the sample was handled to measure these parameters and answer the questions: were the samples from each animal, was the stomach divided, and if so, how and which parts were taken for each determination?
- In the subsection (page 7) "Effect of GGE03 on Mucosal Defensive Factors in EtOH / HCl-treated Rats", the effect of the extract on gastric wall mucus content is not discussed, the effect of which is presented in Figure 3.
- Effect of GGE03 in ethanol-treated rats was improperly described (see: Administration of acidified ethanol significantly reduced PGE2 levels to 29.0 ± 1.0 ng / mg of tissue. This increase was attenuated by pretreatment with 100 mg / kg GGE03 (39.4 ± 0.9 ng / mg of tissue) ”, which in the second sentence is not true, because the effect of ethanol causes a decrease, not an increase, of PGE2 level (see Figure 3).
- Authors mention that increasing 1-dehydro-6- gingerdione may be responsible for the effect of the tested extract (discussion chapter). As there are known works on the common extract of this plant (see: references 21 and 22), it would be worth discussing whether or not other components of this plant are involved.
Minor errors:
- The abstract probably mentions the model of ulcer induction by the use of aspirin by mistake, which was not the case in this study.
- Authors should indicate how many animals were housed in one cage during the experiment.
- Please make a list of abbreviations.
- Typographic artifact (page 7, line 1:“qHistological (….)”).
- The name: normal control (NC) or sham + vehicle group should be used consistently throughout the body text and figures (see: Figures 2-5, page 7).
- Subsection title (page 7): “3. Effect of GGE03 on Mucosal Defensive Factors in EtOH / HCl-treated Rats” should appear under number 3a.
Author Response
Dear Editor, Dear Reviewers,
We would like to submit a revised version of our manuscript entitled ” Antiulcer Activity of Steamed Ginger extract, GGE03, against Ethanol/HCl-induced Gastric Mucosal Injury in Rats” for publication in Molecules. We appreciated the overall positive evaluation of our manuscript by the three reviewers and we would like to thank the reviewers for their efforts and constructive criticism that helped us to improve the quality of our manuscript. Please find below a point by point response to the issues raised by the reviewer. - All changes in the manuscript are marked in yellow highlighting. –
Reviewer #1
Comment 1. Citing item 1 (Liu E.S., Cho C. Relationship between ethanol induced gastritis and gastric ulcer formation in rats. Digestion 2000; 62: 232-239) as a source of information on the epidemiology of peptic ulcer disease is at least inappropriate and misleading. After all, it is devoted to one of the methods of conducting research that mimics the disease on experimental animals. Authors should place here the latest data on this disease from updated, as recent reports.
Response: We agreed with the reviewer’s comments. First, we changed Reference 1 to reflect the latest data on gastric ulcer (World Journal of Gastroenterology. 2013; PMID: 23372356). Second, to discuss oxidative stress and its relationship with gastric ulcer, we changed Reference 7 (Hepatogastroenterology. 2001 May, PMID: 11462918). Furthermore, we cited another Reference 8 (Journal of Hepatology. 2009, PMID:19398236) to support the concept that the alcohol-induced reactive oxygen species related to the several release of pro-inflammatory cytokines. Finally, we revised manuscript and references 11-13 to discuss the side-effect and limitation of commonly used gastric ulcer drugs in revised manuscript (Reference 11: N Engl J Med 1976 Oct, PMID: 958284; Reference 12: JAMA 2013 Dec, PMID: 24327038; Reference 13: Clin Kidney J 2012 Aug, PMID: 25874082).
Comment 2. A similar comment can be made to the introduction section devoted to oxidative stress and its relationship with peptic ulcer disease. The authors cite position 7 (Rezaie A., Parker R.D., Abdollahi M. Oxidative stress and pathogenesis of inflammatory bowel disease: An epiphenomenon or the cause? Dig Dis Sci . 2007; 52: 2015-2021) in it, which not only is a bit older, but also mainly refers to Crohn's disease and ulcerative colitis. At this point, Authors should cite works strictly devoted to the discussed topic
Response: Thanks for your suggestion. Sorry for this mistake. We revised it with an appropriate reference which is cited inside the text. “7. Kountouras J.; Chatzopoulos D.; Zavos C. Reactive oxygen metabolites and upper gastrointestinal diseases. Hepatogastroenterology. 2001, 48, 743-751.”
Comment 3. Also the citation of position 8 (Nian M., Lee P., Khaper N., Liu P. Inflammatory cytokines and postmyocardial infarction remodeling. Circ Res. 2004; 94: 1543-1553) is not very appropriate and does not fully reflect the idea presented in in the sentence (pages 1-2: "Further, alcohol-mediated ROS is a crucial factor in inflammation via increasing formation of pro-inflammatory cytokines, such as tumor necrosis factor-α (TNF-α) and interleukin-1β (IL-1β) , which exacerbate inflammation [8] ”), because to support this thesis there are many other reports dealing with the exact alcohol compound, ROS and the pro-inflammatory processes related to the release of various pro-inflammatory cytokines.
Reponse : Thanks you for your point out our mistake. We corrected it with an appropriate citation. Mandrekar P.; Szabo G. Signalling pathways in alcohol-induced liver inflammation. J. Hepatol. 2009, 50, 1258-1266.
Comment 4. It also seems that the citation of item 11 (Zakaria et al. Gastroprotective activity of chloroform extract of Muntingia calabura and Melastoma malabathricum leaves. Pharm Biol. 2016: 54: 812-826) in relation to the sentence (page 2, paragraph 2: “However, these drugs have side effects, such as gynecomastia, impotence, osteoporotic bone fracture, and deficiencies of iron and magnesium, as well as vitamin B12 hypergastrinemia after discontinuation”) discussing the adverse concerns about drugs H2 receptor antagonists, proton pump inhibitors and antacids is unjustified and not directly related to the idea contained in this fragment of the reviewed publication.
Response: Thank you for your comment. We revised citations to relate with side effect and limitation of primary gastric ulcer drugs. “[11] Hall, W.H. Letter: Breast changes in males on cimetidine. N. Engl. J. Med. 1976, 295, 841.”. “ [12]Lam, J.R.; Schneider, J.L.; Zhao, W.; Corley, D.A. Proton pump inhibitor and histamine 2 receptor antagonist use and vitamin B12 deficiency. JAMA. 2013, 310, 2435-2442.”, “[13] Singh, A.; Ashraf, A. Hypercalcemic crisis induced by calcium carbonate. Clin. Kidney J. 2012, 5, 288-291.”
Comment 5. Many parameters have been measured after inducing simulating ulcer formation in experimental animals (see: Materials and Methods chapter). It seems that Authors should describe exactly how the sample was handled to measure these parameters and answer the questions: were the samples from each animal, was the stomach divided, and if so, how and which parts were taken for each determination?
Response: Thank you for your valuable comments. We exactly described how and which part of the stomach was used for both histological and biochemical analysis (Page 4). “The stomachs from each group were opened along the greater curvature, and thoroughly rinsed with normal saline. A longitudinal section of the gastric tissue was taken from the anterior part of the stomach and fixed in 4% formalin solution for later histological analysis and the remaining stomach sections were stored at -75 °C for later biochemical analysis”
Comment 6. In the subsection (page 7) "Effect of GGE03 on Mucosal Defensive Factors in EtOH / HCl-treated Rats", the effect of the extract on gastric wall mucus content is not discussed, the effect of which is presented in Figure 3.
Response: Thank you for your point out our mistake. We revised our manuscript (page 7). “Rat treated with acidified EtOH showed a significant decrease in the gastric wall mucus compared to the vehicle-treated sham group. Administration of GGE03 100 mg/kg significantly attenuated this decrease (Figure 3a).”
Comment 7. Effect of GGE03 in ethanol-treated rats was improperly described (see: Administration of acidified ethanol significantly reduced PGE2 levels to 29.0 ± 1.0 ng / mg of tissue. This increase was attenuated by pretreatment with 100 mg / kg GGE03 (39.4 ± 0.9 ng / mg of tissue) ”, which in the second sentence is not true, because the effect of ethanol causes a decrease, not an increase, of PGE2 level (see Figure 3).
Response: Thank you for your point out our mistake. We revised statement of PGE2 level (page 7). “This increase -> This reduction”
Comment 8. Authors mention that increasing 1-dehydro-6- gingerdione may be responsible for the effect of the tested extract (discussion chapter). As there are known works on the common extract of this plant (see: references 21 and 22), it would be worth discussing whether or not other components of this plant are involved.
Response: Thank you for your suggestion. We revised our manuscript (page 11). “Lee et al., [56] demonstrated that 1-dehydro-10-gingerdione suppressed NF-κB-regulated gene expression through inhibiting the catalytic activity of cell-free IKKβ in lipopolysaccharides-treated macrophage.”
Minor errors:
- The abstract probably mentions the model of ulcer induction by the use of aspirin by mistake, which was not the case in this study.
- Response: Thank you for your suggestion. It was our mistake. To avoid the reader’s confusion, we deleted that sentence in our revised manuscript (Page 1).
- Authors should indicate how many animals were housed in one cage during the experiment.
- Response: Thank you for your suggestion. We revised our manuscript (2.3 Animal treatment, page 3). “Animals were paired-housed (2 per cage) in a room with controlled temperature and humidity (25 ± 1°C and 55 ± 5%, respectively) and with a 12 h light-dark cycle. ”
- Please make a list of abbreviations.
- Response: Thank you for your suggestion. We make a list of abbreviations (page 11). “NSAIDs, Non-steroidal anti-inflammatory drugs; ROS, Reactive oxygen species; SOD, Superoxide dismutase; GPx, Glutathione peroxidase; CAT, Catalase; TNF-α, Tumor necrosis factor-α; IL-1β, Interleukin-1β; PGs, Prostaglandins; NO, Nitric oxide; NF-κB, nuclear factor κ-activated B; EtOH, Ethanol; GGE, Golden ginger; MDA, malondialdehyde; TBA, thiobarbituric acid; MPO, Myeloperoxidase; TBS/T, Tris-buffered saline; GAPDH, glyceraldehyde-3-phosphate dehydrogenase; COX, cyclooxygenase; NOS nitric oxide synthase; GSH, glutathione.”
- Typographic artifact (page 7, line 1:“qHistological (….)”).
- Response: It was our mistake. In our revised manuscript, we removed extra ‘q’ in Page 7.
- The name: normal control (NC) or sham + vehicle group should be used consistently throughout the body text and figures (see: Figures 2-5, page 7).
- Response: thanks you for your point out our mistake. To avoid reader confusion, we changed from ‘Normal Control’ group to ‘sham + vehicle’ group in our revised manuscript.
- Subsection title (page 7): “3. Effect of GGE03 on Mucosal Defensive Factors in EtOH / HCl-treated Rats” should appear under number 3a.
- Response: thanks you for your point out our mistake. We revised subsection title. “3.2 Effect of GGE03 on Mucosal Defensive Factors in EtOH / HCl-treated Rats”

Reviewer 2 Report
This is a straight forward descriptive manuscript in which the authors demonstrate the preventive effect of steamed ginger extract against gastric ulcer induced by acid-alcohol.
The reviewer has several comments:
In the title, GGE03 should be removed.
In the introduction:
The authors should indicate in the text when they first use the abbreviation (GGE) what does each letter and number refer to? Golden Ginger Extract (GGE), how about “03” ??
Some references are not appropriate, for example #1. The authors should revise/double check references in introduction and discussion and make sure they reflect the statements written.
In the methods:
It will be interesting if the investigators have added one group to demonstrate that ginger extract prepared in the same way as previously published (un-steamed) is less effective on gastric ulcer model than the seamed preparation. Otherwise, this would support what is indicated in the introduction.
It is not clear whether a stomach of each mouse was processed for the different assays or for each assay they used the whole stomach. If one stomach is used for different assays, it has to be indicated which region of the stomach was used for each assay. Or perhaps a scheme could be added to label the assays done for each region of the stomach.
Page 2, the sentence starting in line 3 needs to be corrected (occur: occurs; … also anther verb is missing). There are many other typos, the authors should carefully revise the text.
Page 2, line 8, no need for “after discontinuation”..
Scheme 1 should be expanded to include the second part of the method (elution and quantification of 1-dehydro-6-gingerdione) and also include some details (time, temp,.. chemicals' names…) so that the scheme becomes more meaningful and can be followed with a little need to refer to the text.
Page 3, last paragraph, the authors should indicate in the text how they induced gastric ulcer; it’s not enough to give a reference!!
Page 4, paragraph 3, last sentence, the authors should briefly indicate the method that they referred to and.. what are the slight modifications?
Page 5, paragraph 3, “cytosolic” should be replaced with “cytoplasmic” and also in the whole text of the manuscript. Paragraph 4, first line does not read properly; it requires corrections.. It also sounds like a repetition of what is mentioned in the previous paragraph!
In the results:
Page 7, remove extra ‘q’ at the beginning of the page!!
Page 8: use full name for Malondialdehyde when first mentioned. Then use (MDA) in the rest of the text
Figure 2, the resolution is very poor and should be improved
In the discussion: at the end of the 2nd paragraph, there is no need for the sentence about NO (NO, a powerful vasodilator, regulates bicarbonate secretion, blood flow, ….) or it can be added in the next paragraph.
Author Response
Reviewer #2.
This is a straight forward descriptive manuscript in which the authors demonstrate the preventive effect of steamed ginger extract against gastric ulcer induced by acid-alcohol.
The reviewer has several comments:
- In the title, We changed title as your comment “GGE03 removed”
- In the introduction: The authors should indicate in the text when they first use the abbreviation (GGE) what does each letter and number refer to? Golden Ginger Extract (GGE), how about “03”
- Response: Thank you for nice comment. We corrected “GGE03”
- Some references are not appropriate, for example #1. The authors should revise/double check references in introduction and discussion and make sure they reflect the statements written.
- Response: Thank you for your valuable comments. We already insert all correct reference in the manuscript and reference sections.
- In the methods: It will be interesting if the investigators have added one group to demonstrate that ginger extract prepared in the same way as previously published (un-steamed) is less effective on gastric ulcer model than the seamed preparation. Otherwise, this would support what is indicated in the introduction.
- Response: Thank you for your valuable suggestion. We do not have un-steamed stock.
- It is not clear whether a stomach of each mouse was processed for the different assays or for each assay they used the whole stomach. If one stomach is used for different assays, it has to be indicated which region of the stomach was used for each assay. Or perhaps a scheme could be added to label the assays done for each region of the stomach.
- Response: Thank you for your point out of our mistake. We revised how and which part of the stomach was used for both histological and biochemical analysis at materials and methods part (Page 4).
- Page 2, the sentence starting in line 3 needs to be corrected (occur: occurs; … also anther verb is missing). There are many other typos, the authors should carefully revise the text.
- Response: Thank you for your point out of our mistake. We revised several typographic errors in our manuscript.
- Page 2, line 8, no need for “after discontinuation”.
- Response: Thank you for your suggestion. We removed the unnecessary parts.
- Scheme 1 should be expanded to include the second part of the method (elution and quantification of 1-dehydro-6-gingerdione) and also include some details (time, temp,.. chemicals' names…) so that the scheme becomes more meaningful and can be followed with a little need to refer to the text.
- Response: Thank you for your point out of our mistake. We revised Scheme 1.
- Page 3, last paragraph, the authors should indicate in the text how they induced gastric ulcer; it’s not enough to give a reference!!
- Response: Thank you for your valuable comment. We describe the detailed method of acute gastric ulcer model in our revised manuscript. “Acute gastric ulcers were induced by EtOH/HCl treatment following a previous report [28, 29]. In our preliminary experiment, after 24 h of fasting, rats were orally administered GGE03 (10, 30, 100, and 300 mg/kg), vehicle (normal saline) as a sham, or indomethacin (100 mg/kg) as a positive control, 1 h before receiving 1 mL of 150 mM HCl in 60% ethanol (p.o.).”
- Page 4, paragraph 3, last sentence, the authors should briefly indicate the method that they referred to and.. what are the slight modifications?
- Response: Thank you for your comment. We applied double blind test (Specimens were scored by two independent observers blinded to the experimental setup under light microscopy using a scoring system).
- Page 5, paragraph 3, “cytosolic” should be replaced with “cytoplasmic” and also in the whole text of the manuscript. Paragraph 4, first line does not read properly; it requires corrections.. It also sounds like a repetition of what is mentioned in the previous paragraph!
- Response: Thank you for your valuable comment. We replaced “cytosolic” to “cytoplasmic” in our revised manuscript.
- In the results: Page 7, remove extra ‘q’ at the beginning of the page!!
- Response: It was our mistake. In our revised manuscript, we removed extra ‘q’ in Page 7.
- Page 8: use full name for Malondialdehyde when first mentioned. Then use (MDA) in the rest of the text
- Response: Thank you for your point out of our mistake. We already revised it in manuscript.
- Figure 2, the resolution is very poor and should be improved
- Response: Thank you for your comment. We revised it.
- In the discussion: at the end of the 2nd paragraph, there is no need for the sentence about NO (NO, a powerful vasodilator, regulates bicarbonate secretion, blood flow, ….) or it can be added in the next paragraph.
- Response: Thank you for your comment. We added that sentences in the next paragraph in our revised manuscript (Page 10)
